# Multifunctional Polymeric Micelles for Cancer Therapy

**DOI:** 10.3390/polym14224839

**Published:** 2022-11-10

**Authors:** Geun-Woo Jin, N. Sanoj Rejinold, Jin-Ho Choy

**Affiliations:** 1Intelligent Nanohybrid Materials Laboratory (INML), Institute of Tissue Regeneration Engineering (ITREN), Dankook University, Cheonan 31116, Korea; 2R & D Center, CnPharm Co., Ltd., Seoul 03759, Korea; 3Division of Natural Sciences, The National Academy of Sciences, Seoul 06579, Korea; 4Department of Pre-Medical Course, College of Medicine, Dankook University, Cheonan 31116, Korea; 5International Research Frontier Initiative (IRFI), Institute of Innovative Research, Tokyo Institute of Technology, Yokohama 226-8503, Japan

**Keywords:** polymeric micelle, multifunctionality, cancer therapy

## Abstract

Polymeric micelles, nanosized assemblies of amphiphilic polymers with a core–shell architecture, have been used as carriers for various therapeutic compounds. They have gained attention due to specific properties such as their capacity to solubilize poorly water-soluble drugs, biocompatibility, and the ability to accumulate in tumor via enhanced permeability and retention (EPR). Moreover, additional functionality can be provided to the micelles by a further modification. For example, micelle surface modification with targeting ligands allows a specific targeting and enhanced tumor accumulation. The introduction of stimuli-sensitive groups leads to the drug’s release in response to environment change. This review highlights the progress in the development of multifunctional polymeric micelles in the field of cancer therapy. This review will also cover some examples of multifunctional polymeric micelles that are applied for tumor imaging and theragnosis.

## 1. Introduction

Advancements in nanoscience and nanotechnology have proposed innovative ways for the treatment of diseases, especially cancer, that pose a significant challenge for clinicians [1,2,3,4]. Researchers are currently investigating a variety of nanosized drug-delivery systems for enhanced delivery of anticancer drugs, including small-molecule drugs and genes. These nanomedicines can provide effective ways to successfully overcome many drawbacks of anticancer drugs associated with poor solubility, poor pharmacokinetics (PK), non-selectivity, and toxicity. The advantages achieved by the nanomedicines studied to date are the ability to solubilize hydrophobic drugs, improved pharmacokinetics (PK), controlled drug release, biocompatibility, targetability, and responsiveness to external stimuli, enabling triggered drug release, thereby improving the overall efficacy [5,6,7,8,9,10].

Previously, specific reviews have been published on polymeric micelles for cancer therapy. For example, the review by Kaur et al., (2022) specifically focused on the clinical challenges of polymer-based micelles [11]. Similarly, Junnuthala et al., (2022) reviewed applications of micelles specifically for breast cancer therapy [12]. Similar studies have been reported earlier as well. Exemplifying such studies is the review of polymeric micelles for cancer therapy published by A Varela-Moreira et al., (2017) [13]. Here we review only the latest updates on polymeric micelles for cancer therapy.

Therefore, the present review focuses on the new developments on polymeric micelles, which have a core–shell nanostructure formed by self-assembly of amphiphilic polymers. Adding to the tumor-targeting property of polymeric micelles via the EPR effect, various modifications of polymeric micelles enabling active targeting and stimuli-sensitive release are discussed. Finally, we discuss multifunctional polymeric micelles, which are applied for tumor imaging and theragnosis.

## 2. Polymeric Micelles

Before discussing multifunctional polymeric micelles, the basic property of polymeric micelles will be reviewed in this section. Polymeric micelles are nanosized constructions formed from the self-assembly of amphiphilic copolymers with both hydrophobic and hydrophilic compartments in their structure. The self-assembly above critical micelle concentration (CMC) is an entropically favored phenomenon. During the process, the hydrophobic compartment of the copolymer forms the core of polymeric micelles, whereas the hydrophilic compartment forms the shell.

Various amphiphilic di- or triblock copolymers have been used to form the polymeric micelles. The widely adopted hydrophilic compartment for the copolymer is poly(ethylene glycol) (PEG). PEG is hydrophilic, nonionic, biocompatible, and inert to many organic compounds. The antifouling property of PEG, which is related to steric hindrance effects, is known to contribute to the protection of drugs from enzymatic degradation. In some cases, poly(N-isopropylacrylamide) (PNIPAAm) was used as a hydrophilic compartment in a copolymer to introduce a thermos-sensitive property to micelles [14]. Unlike the hydrophilic compartment, a variety of polymers have been utilized as hydrophobic compartments in copolymers. Polymers such as poly(propylene glycol) (PPG) [15], poly(ε-caprolactone) (PCL) [16], and poly(L-lactide) (PLA) [17] are representative examples of the polymers used as hydrophobic compartments. Some polymers are hydrophilic under specific conditions but can form the micelle core. Poly(L-histidine) (PHis) [18] has been studied due to its pH sensitivity. The imidazole moiety in histidine gives an amphoteric property to pHis through protonation/deprotonation. Phis is hydrophobic at physiological pH but protonated, exhibiting a positive charge at the acidic pH below 6. Poly(L-aspartic acid) (PAsp) has been used to form another special kind of micelle. Kataoka et al. reported a type of “polyion complex (PIC) micelle” driven by electrostatic interactions in an aqueous solution of negatively charged block, PAsp with cationic cargos [19]. 

Polymeric micelles have several features suitable for drug-delivery applications. Most anticancer drugs have low solubility in water due to their polycyclic structure, and their clinical uses are limited [20]. Furthermore, the low solubility coupled with drug metabolic degradation results in poor bioavailability and low systemic drug exposure [21]. To solubilize the hydrophobic anticancer drugs, formulation vehicles have been used. However, the use of the formulation vehicles has been associated with hypersensitivity reactions such as hyperlipidaemia, erythrocyte aggregation, and peripheral neuropathy [22]. The hydrophobic core of polymeric micelles can act as a reservoir for hydrophobic drugs. Incorporation of these drugs in polymeric micelles avoids the use of formulation vehicles and resulting toxicity [23]. Using polymeric micelles as delivery carriers for anticancer drugs has several advantages over conventional chemotherapy. The size of polymeric micelles is the special feature allowing them to circulate in the blood stream for a long time by evading their clearance in the liver and kidneys. Their size is small enough to evade the mononuclear phagocytic system (MPS) in the liver. Also, their size is large, and they are precluded from renal clearance [24,25]. 

Some of the recent polymeric micelles and their physicochemical characteristics and applications are listed in Table 1.

## 3. Polymeric Micelles with Passive Targeting Function

Longer circulation allows the polymeric micelles to be accumulated in a tumor. The accumulation has been known to be preceded by an enhanced permeability and retention (EPR) effect, which is the basis of passive targeting of polymeric micelles. Growing tumors require a high amount of nutrients and oxygen, and tumors drive blood vessel growth. Tumor vasculatures grow fast, and vascular density tends to decrease as tumors grow. The resulting structural abnormality makes the tumor blood vessels highly permeable [39]. As a result, polymeric micelles can penetrate through the leakage of blood vessels and be retained in tumor tissue. This phenomenon was first reported by Matsumura and Maeda in 1986 and was named the EPR effect [40]. The cutoff size of tumor blood vessel leakage determines the diffusion of polymeric micelles in a tumor, and the cutoff size is suggested to be between 400 and 600 nm in diameter [41].

Polymeric micelles were used to deliver hydrophobic anticancer drugs to tumors using a passive targeting effect. Yu et al. designed a PEG-camptothecin conjugate (PEG-CPT) and evaluated its antitumor effect in a xenograft model of human colon cancer. The results showed that PEG-CPT provided improved biodistribution and increased the drug’s stability, offering better tumor uptake with reduced side effects. PEG-CPT did not induce apoptosis in the liver and kidneys. However, PEG-CPT enhanced apoptosis induction in the tumor [42]. Notably, PEG-PLA copolymer-based micelles were approved in the Republic of Korea in 2007. Genexol^®^-PM (Samyang Co., Seoul, Korea) is a polymeric micellar formulation of paclitaxel. Genexol^®^-PM showed greater anticancer activity and higher tumor tissue concentration compared to paclitaxel [43]. Genexol^®^-PM was approved for the treatment of breast, lung, and ovarian cancers. 

Although passive targeting has been proven useful clinically, it has drawbacks. The major drawbacks of passive targeting are associated with the inaccuracy in targeting the tumor. EPR-mediated passive targeting is considerably dependent on vascular permeability. However, a tumor is induced by not only the outgrowth of the cells but also epigenetic mutation caused by physical and biological signals in a tumor’s microenvironment (TME) [44]. The heterogeneity in TME, composed of various vascular conditions, contributes to the heterogeneity in EPR-mediated accumulation. The heterogeneity includes various receptor expression and interstitial fluid pressure as well as vascular permeability. For this reason, the extent of tumor accumulation varies between tumor types. According to previous studies, the vascular cutoff sizes were various between tumors, and the vasculature exhibits different porosities and permeability, even in the same tumor [45,46]. 

There have been new micelle systems reported recently for multifaceted applications. However, the passive EPR-assisted cancer therapy of micelles is still challenging and could be least specific to many cancers. Therefore, specific targeting functionalities should be done on such micelle structures to impart better therapeutic outcomes.

## 4. Polymeric Micelles with Active Targeting Function

As discussed in the previous section, only passive targeting depending on the EPR effect cannot fully guarantee the effective delivery of polymeric micelles to tumors. Additional approaches are required for robust tumor targeting. Cancer cells and tumor vasculature frequently exhibit enhanced expression of certain antigens or receptors compared to normal cells and tissues. Actively targeting polymeric micelles exploits this feature of cancer cells for selective accumulation of anticancer drugs in the tumor tissue. In the preparation of polymeric micelles for active targeting, the polymeric micelle surface is chemically modified by targeting ligands having a strong binding for antigens or receptors over-expressed on cancer cells. To date, various ligands have been investigated for active targeting of polymeric micelles. 

Antibodies have been mostly adopted as targeting ligands for polymeric micelles, due to their diversity and specificity. Jin et al. prepared PEG-PLGA micelles conjugated with bivalent fragment HAb18 F(ab′)_2_, which binds to hepatocellular carcinoma (HCC) cells with high affinity. The cancer cells (Huh7 and HepG2) incubated with targeted micelles (DOX-loaded HAb18 F(ab′)2-PEG-PLGA micelles) showed nearly 1.5 times higher cellular uptake compared to nontargeted micelles (DOX-loaded PEG-PLGA micelles). DOX-loaded targeted micelles showed higher toxicity than nontargeted micelles in a cell study using HepG2 cells. The animal study using xenograft nude mice bearing HepG2 cells shows that targeted micelles were distributed to the tumor, while the accumulation in the brain, heart, lungs, liver, spleen and kidneys was low. Furthermore, targeted micelles showed significant suppressed tumor growth compared to free DOX and DOX-loaded PEG-PKGA. The calculated inhibition rates of tumor growth (%) were 39.8%, 50.2%, and 63.9% for free DOX, DOX loaded PEG-PLGA, and targeted micelle-treated groups, respectively [47].

Peptides have been used as targeting ligands since their immunogenicity is lower compared to proteins [48]—for example, kNGR (Asn-Gly-Arg) peptide-functionalized lipid–polymer-based NPs for the targeted delivery of Paclitaxel (PTX). The micelle backbone was based on PLGA–lecithin–PEG, whereas the peptide conjugation will boost the CD13 targeting property of the NPs (Figure 1). Approximately 200-nm-sized micelles were made by a nanoprecipitation method and were found to be significantly effective in terms of higher apoptosis and cellular localization on HT-1080 (Human fibrosarcoma cell line). The in vivo results demonstrated tumor inhibition of ~ 60% on an HT-1080 tumor model by these micelle formulations. The experimental outcome assures that such peptide-tagged micelles formulations would be highly effective against various solid malignant tissues with CD13 overexpression [49].

In general, drug delivery to brain tumors are difficult to due to the BBB (blood–brain barrier) [50,51,52,53]. To overcome such barriers, multifunctional fatty acid grafted chitosan micelles have been made for pVGF delivery to the brain and have been found to be effective by an intranasal administration. The as-made hemo- and biocompatible, 200-nm-sized cationic micelles were evaluated by two routes of administration (intranasal and intravenous). The chitosan micelles were modified by the mannose, TAT peptide, and oleic acid and were found to significantly enhance VGF expression in primary astrocytes and neurons and were better when administered in an intranasal route. Both mannose and TAT ligand tags were able to enhance pVGF localization to the brain [54].

Similarly, CD44-targeted amphiphilic micelles (~154 nm) based on polyethylene glycol-block-hydroxyethyl starch-block-poly (L-lactic acid) (CD44p-conjugated PEG-*b*-HES-*b*-PLA) were developed by a self-assembly technique for improving the solubility of a hydrophobic Emodin (Emo) drug for triple negative MDA-MB-231 cells. The cellular uptake and efficacy test were conducted on MDA-MB-231 cells, showing these Emo-loaded polymer micelles could be a potential nanomedicine for CD44-targeted chemotherapy [55].

The α_v_β_3_ integrin receptors have been reported to overexpress in many tumors [56,57,58]. Targeted drug delivery to such receptors can further improve the anticancer therapy very effectively. A selective α_v_β_3_ integrin-specific peptide ((Cyclo(Arg-Gly-Asp-D-Phe-Lys); abbreviated as c(RGDfK)) was functionalized on camptothecin (CPT). The as-made prodrug can make self-assembled NPs for precise targeting with enhanced tumor-homing capability and altogether improved anticancer efficacy as well. In vivo metabolic analysis demonstrated enhanced blood circulation for CPT by the NPs. While the control CPT can be rapidly cleared out from the body, the peptide-targeted NPs were selectively accumulated in tumor tissues, indicating that the cRGD-tagged micelles can effectively target the tumor tissues and could be useful for many other types of solid tumors as well [59].

HER2-targeted multifunctional stable micellar formulation was found to be very effective on breast cancer cells. The well-stable, pH-dependent biodegradable micelles were made by an acid-responsive cross-linking through RAFT polymerization, then a maleimide–DOX was attached to the pyridyl-disulfide-modified micelles. The developed polymers then coupled with a peptide that can bind HER2-receptors. The efficacy was revealed by experimenting on two cell lines, such as MCF-10A (HER2-negative) and SKBR-3 (HER2-positive). The specificity and selective targeting thereby enhanced apoptosis and was determined to be more on HER2-positive cells [60].

Similarly, folic-acid (FA)-targeted, pH-sensitive, cell-membrane-mimicking mixed micelles were made for DOX delivery. The mixed 150-nm-sized mixed micelles, composed of P(DMAEMA-*co*-MaPCL) (PCD) and FA-P(MPC-*co*-MaPCL) (PMCF), were made by a dialysis method (Figure 2). The HeLa and MCF-7 tumor cells were significantly suppressed after being treated with these micelles, suggesting that these nanomicelles could be beneficial for FA-targeted therapy for various tumors [61].

Since phenylboronic acid (PBA) has a targeting ability towards sialic acid (SA), which is overexpressed in tumors, PBA-modified micelles would be highly effective for cancer therapy. Such an idea was established on poly(maleic anhydride) by conjugating not only PBA and F127, but also ethanolamine for doxorubicin (DOX) delivery. DOX release was acid-sensitive, and increased polymer concentration had enhanced tumor targeting, as proven on HepG2, with higher content of SA-containing glycosphingolipids than that of anti-B16. The in vivo studies on the activity of B16-bearing mice showed that the DOX-loaded micelles could inhibit the tumor growth and were safer than free DOX. Thus, the PBA-modified nanopolymer micelles have potential biomedical applications due to their nanostructure and tumor-targeting ability [38].

The recent studies indicated tremendous progress on active targeted polymer micelles for cancer therapy. However, in most of the cases, detailed toxicity and safety studies are not specifically reported.

## 5. Polymeric Micelles with Stimuli-Responsive Function and Their Applications in Photothermal Therapy

The tumor-selectivity of polymeric micelles can be strengthened by adding a stimuli-responsive property. Polymeric micelles can be engineered to respond to various stimuli and release anticancer drugs. Especially, stimuli such as low pH, hypoxia, temperature, electromagnetic waves (NIR, RF(radio frequency) etc.), and cancer-specific enzymes are commonly found in tumors. The low pH in tumors has been widely exploited to design stimuli-responsive polymeric micelles. The acidic pH in tumor results from the excess lactic acid produced by cancer cells via altered metabolism. The tumoral pH is 6.5 compared to 7.4 in the normal tissues. The decreased pH is known to promote multiple processes such as angiogenesis, metastasis, and cancer invasion [62].

Kim et al. synthesized cross-linked polymer micelles by using PEG-conjugated poly(methacrylic acid) (PEG-PMA) copolymers as templates. The cross-linking was achieved by reacting PEO-PMA block copolymers with an oppositely charged metal cation, Ca^2+^. The DOX-loaded polymeric micelles exhibited pH-sensitive release behavior with accelerated release of DOX in an acidic environment. Cytotoxic activity of DOX-loaded PEG-PMA micelles was evaluated against human ovarian cancer cells (A2780) and was compared to Doxil^®^. DOX-loaded PEG-PMA micelles showed lower cytotoxicity than free DOX, which showed a higher IC50 value compared to Doxil^®^. This result suggests that the sustained DOX release from the micelles results in the reduction in the cytotoxicity of DOX [63].

A thermosensitive PCL micelle was conjugated with mesoporous silica nanoparticle (MSN) by disulfide link. The DOX was loaded in both micelles and MSN, enabling a pH/redox/temperature-sensitive DOX@MSN-S_2_-F127-PCL@DOX, abbreviated as DMSFPD. When the temperature was ~40 °C, F127-PCL250 (FP250), the micelle could dissociate from MSN with GSH stimuli, and the shrunken FP250 micelle enabled a sudden DOX release from both MSN and the micelle, causing cytotoxic effects on various cancer cells in vitro [64].

Compared to the single use of photodynamic therapy (PDT), it was suggested that the combination of hypoxia-sensitive therapy could improve its overall efficacy. To realize such a combined therapy, it is important to control the drug release and activate the prodrug in a safe manner. Such a system was made using an NIR-sensitive nanomicelle, ((EGylated cypate (pCy) and mPEG-polylactic acid (mPEG2k-PLA2k)) with a bioreductive prodrug (tirapazamine, TPZ). The whole system was designated as (TPZ@pCy) and made by a nanoprecipitation technique against metastatic breast cancer. Intravenously administered TPZ@pCy accumulated predominantly in the tumor, which could ablate the tumor and lung metastasis with no severe toxicity to other major organs (Figure 3) [65]. 

Ruthenium (Ru) complexes are known for anticancer therapy. However, the selectivity towards cancer cells is very difficult in addition to the low production of ROS for PDT. Such limitations could be overcome by micellar modification of the Ru complex. For example, a biodegradable Ru-containing polymer (poly(DCARu)) was developed for integrating two therapeutic molecules (the drug and the Ru complex) by conjugating a diblock copolymer (MPEG-*b*-PMCC) containing hydrophilic poly(ethylene glycol) and cyano-functionalized polycarbonate. Upon NIR irradiation, these micelles were able to inhibit the tumor growth effectively in vivo [34]. In another work, 2-pyridone-bearing BODIPY photosensitizer was linked to polyethyleneglycol-*b*-poly(aspartic acid) to form a photosensitizer-_1_O^2^ generation, storage/release agent dual-loading system (PEG-*b*-PAsp-BODIPY) for cancer therapy [35]. A host–guest interaction of benzimidazole-terminated PHEMA-*g*-(PCL-BM) and β-CD-star-PMAA-*b*-PNIPAM enabled an 80 nm micelle for thermosensitive drug delivery of DOX. The lowest critical solution temperature (LCST) was between 40 and 41 °C and the DOX-loaded micelles displayed excellent anticancer activity compared to the innate DOX [36]. A 120-nm-sized, IR780-loaded, PLA-based micelle showed selective toxicity to MDA-MBA-231 cells under laser irradiation, suggesting its potential as a multifunctional theranostic agent for breast cancer treatment with increased cellular uptake, PDT, and more reliable tracking in cell-image studies [37].

PLA-based micelles were developed for Ru loading for enhanced cancer therapy. The micelles are composed of (MPEG-SS-PMLA) of poly(ethylene glycol) and phenyl-functionalized poly(lactic acid) with a disulfide bond. The MPEG-SS-PMLA had ~83% Ru content, thanks to the π-π bonding among the phenyl ring and Ru complex. Under the GSH condition of 10 mmol/L, 70% of Ru was released to cause apoptosis on MCF-7 through PDT (Figure 4) [27].

It is well known that the synergy between PTT (photothermal therapy) and chemotherapy could improve overall anticancer efficacy. However, developing stable photosensitizer-loaded micelles is always challenging. A biodegradable hybrid micelles (HMs) nano system was developed for the for co-delivery of paclitaxel (PTX) and IR825. The hybrid micelle (PTX/IR825-TAT HMs) was self-assembled through hydrophobic interactions between polyethyleneimine-polycaprolactone (PEI-PCL) and TAT peptide-modified-1,2-distearoyl-sn-glycero-3-phosphoethanolamine-N-(polyethylene glycol)5000 (TAT-PEG-DSPE). In vivo antitumor results showed that PTX/IR825-TAT HMs can selectively enter tumor tissue, enabling chemophototherapy to be effective [66].

Similarly, indocyanin green (ICG) was loaded into amphiphilic diblock polymer poly(ethylene glycol)–poly(l-tyrosine-^125^I) for PTT and multifaceted imaging such as fluorescence, SPECT (ingle photon emission computed tomography), and PAI, as well [67].

Even though there are various reports on stimuli-responsive polymer micelles, their clinical applicability is still challenging due to the lack of long-term toxicity analyses.

## 6. Polymeric Micelles with Imaging and Theranostic Functions and Their Applications in Photodynamic Therapy

Multifunctional polymeric micelles have been extensively investigated for better tumor-specific delivery of anticancer drugs. A number of combinations of active targeting and stimuli-sensitive properties have been explored to achieve multifunctionality. Most of the polymer-drug conjugates (PDCs) have been made by click chemistry, which is catalyzed by a copper(I) reaction. In 2022, the Nobel Prize—which was shared by Carolyn R. Bertozzi (Stanford University, CA, USA), Morten Meldal (University of Copenhagen, Denmark), and K. Barry Sharpless (Scripps Research, La Jolla, CA, USA) for their contributions along with bio-orthogonal chemistry [68,69,70,71]—recognized click chemistry.

Sone et al. developed a targeting-clickable and tumor-cleavable polyurethane micelle for multifunctional delivery of DOX. The multifunctional polymer was prepared by the conjugation of folic acid (FA) ligand via click chemistry. In the tumor site, PEG is detached in response to tumoral pH by the cleavage of the benzoic–imine bond. Then, intracellular drug release is triggered by the disulfide bond cleavage in response to glutathione (GSH). In the flow cytometry study, FA-conjugated micelles showed an enhanced cellular uptake compared to nontargeting micelles. FA-conjugated micelles efficiently delivered DOX in the cells and DOX fluorescence was observed mainly in the cell cytoplasm. The cytotoxicity of DOX-loaded FA-conjugated micelles were compared with DOX-loaded micelles and free DOX. The study result showed that free DOX was more toxic than DOX-loaded micelles, since free DOX was internalized to the cell nucleus faster than DOX released from DOX-loaded micelles. However, FA-conjugated micelles exhibited a higher cytotoxicity compared to non-targeted micelles, which is consistent with its enhanced cellular uptake observed in flow cytometry [72].

A smart micellar multifunctional nanosystem was made for hepatocellular carcinoma (HCC) by sialic acid (SA) modification on the micelles. The upper critical solution temperature (UCST) of the micelle, i.e., (sialic acid-polyethylene glycol)-poly(acrylamide-co-acrylonitrile), SA-PEG-p(AAm-*co*-AN)), was ~43 °C. The developed micelles were used for DOX and Gd-CuS nanoparticles (Gd-CuS NPs) encapsulation for chemophotothermal treatment of HCC guided by magnetic resonance (MR)/photoacoustic (PA) dual-mode imaging. The resultant SA-PEG-p(AAm-*co*-AN)/DOX/Gd-CuS (SPDG) had superior potential in MR/PA dual-mode imaging-guided chemophotothermal treatment [73].

Click chemistry was used for creating self-assembled polymer-drug conjugates (PDCs). Two amphiphilic PDCs were modified with PEG for loading 5-fluorouracil and coumarin, providing a unique opportunity to develop promising codelivery carriers for synergistic cancer therapy. The micelles showed anticancer efficacy on PANC-1 and BxPC-3, confirming that both coumarin and 5-fluorouracil retain their anticancer properties after conjugation with PEG [74].

Inflammatory bowel diseases ((ulcerative colitis (UC)) can be treated successfully by siRNAs by oral route. However, oral administration of naked siRNA is challenging, as it can easily be damaged by the harsh gastrointestinal tract. A nanocarrier of micelles based on MPEG-PCL-CH2R4H2C was used for oral delivery of siRNA to treat a dextran sodium sulfate-induced UC model [75]. The preliminary results are promising and could be extended for colorectal cancers as well. For example, capecitabine (CAP) was loaded with pH-responsive beta-cyclodextrin (CD)-based micelles for colon cancer therapy [76]. Similarly, there have been wide varieties of multifunctional polymer micelles for magnetic hyperthermia application as well [77]. Such a study by Cheng et al., (2022) reported multifunctional micellar theranostic systems for cancer therapy. The hybrids consist of glucose and TEMPO (2,2,6,6-tetramethylpiperidin-1-yl)oxyl) at the distal ends of PEO-*b*-PLLA block copolymer (glucose-PEO-*b*-PLLA-TEMPO), which can encapsulate CUDC101 and IR780 effectively. The specific core–shell rod structure formed by the designed copolymer gives TEMPO radicals excellent stability against reduction-induced magnetic resonance imaging (MRI) silence. The glucose enables tumor targeting, whereas the TEMPO can have trimodal imaging via MRI, photoacoustic imaging, and fluorescence. Efficient delivery of CUDC101 and IR780 is achieved to synergize the antitumor immune activation through IR780-mediated photodynamic therapy (PDT) and CUDC101-triggered CD47 inhibition, showing M1 phenotype polarization of tumor-associated macrophages (TAMs). More intriguingly, this study demonstrates that PDT-stimulated p53 can also re-educate TAMs, providing a combined strategy of using dual tumor microenvironment remodeling to achieve the synergistic effect in the transition from a cold immunosuppressive to a hot immunoresponsive tumor microenvironment [78].

Recently, there have been huge improvements in theranostic approaches using polymer-based micelles; however, there must still be careful analysis of their long-term applicability to improve cancer therapy.

## 7. Clinical Trials

Even though there are many reported studies related to polymer micelles in the clinical trials, some of the recent ones are listed in Table 2. It was found that most of such studies are focused on either docetaxel and paclitaxel drug formulations as passively targeted formulations. As per Table 2, the only formulation with active targeting under clinical trial is the one with cetuximab/5-FU (5-flouro uracil) combined with the previous clinical trial of NC-6004.

The NC-6004 clinical-drug-based polymer micelle nanotechnique has been studied. This specific drug has a particle size of ~30 nm and the basic reaction chemistry is based on a polymer–metal complex between polyethylene glycol poly (glutamic acid) block copolymers (PEG-P(Glu)) [79]. This ultrafine-sized clinical drug can be tumor-homed through the EPR path [40]; additionally, it was reported that NC-6004 could be in blood circulation longer than its original intact form (conventional cisplatin), which has a short half-life, thanks to the PEG shell on the NC-6004, which can sustain blood circulation significantly higher than the conventional drug [78]. Such beneficial properties of NC could give rise to enhanced pharmacokinetic profile with increased AUC while decreasing the C_max_, availing a safer therapeutic window than the intact cisplatin. Laboratory experiments confirmed the safety of NC-6004 in terms of lowering the nephrotoxicity and neurotoxicity more than the intact cisplatin at an equivalent dose [80]. Several other reported clinical trials using polymer micelle particles are shown in Table 2.

## 8. Conclusions

In this review, we have discussed various examples that cover a wide range of polymeric micelles applied in the field of anticancer drug delivery. Passive targeting with polymeric micelles has great potential; however, TME (tumor microenvironment) heterogeneity limits the potential. To overcome this, various functionalities were employed in polymeric micelles. The block copolymers forming micelles can be chemically modified, facilitating the employment of various functionalities. Due to this, polymeric micelles could be further developed for active targeting and triggered drug release in response to special cues provided by the tumor microenvironment. However, the design of functional polymeric micelles should consider various factors, including targeting, drug release, therapeutic efficacy, biodistribution, and systemic toxicity. In spite of rapid benchside developments, the translation of polymeric micelle to the bedside has been less impressive. Only a few micelle-based drug delivery systems have been approved by the Food and Drug Administration (FDA). The slow bench-to-bedside translation is because of several challenges that need careful research. The main reason is the lack of preclinical tissue culture systems that mimic biological conditions to predict the therapeutic activity and targeting efficiency performance of polymeric micelles. Thus, new evaluation tools should be further developed to accelerate the clinical translation of polymeric micelles. With the development of multifunctional polymeric micelles, the structure of the polymeric micelles becomes more complex, leading to difficulties in reproducible synthesis and scale-up. Impediments to clinical translation can originate from the challenge in developing a robust manufacturing process meeting the regulatory requirements.

In spite of the hurdles mentioned above, more well-designed multifunctional micelles will be investigated, and there will be more accumulated cases translated to clinical trials. In addition, a micellar drug-free therapeutic area should also be well utilized with new functionalities in the coming days. The multidisciplinary field related to chemistry, medicinal science and clinical research holds the promise of delivering breakthroughs in polymeric micelle-based drug delivery, facilitating their move from concept to clinical fields. However, there is long way to go for such establishments because of the lack of detailed toxicity assessments on novel polymer micelles. Therefore, newly developed micelles should be thoroughly checked for toxicity and long-term human applications.

## Figures and Tables

**Figure 1 polymers-14-04839-f001:**
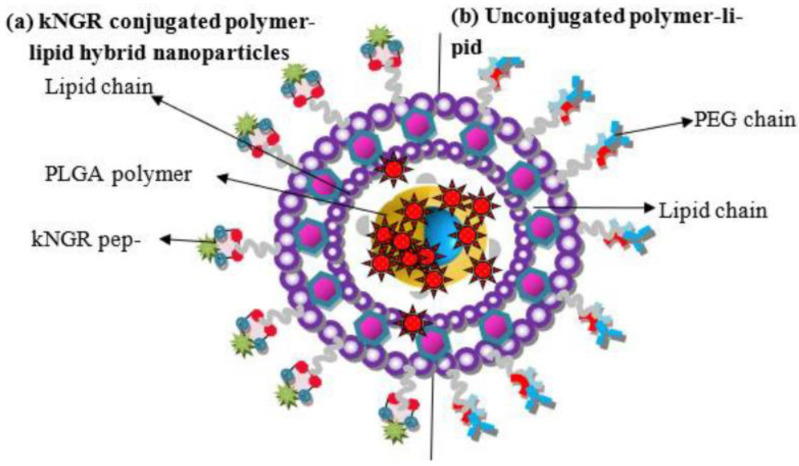
Structure of polymer–lipid hybrid NPs with schematic comparison of (**a**) kNGR-conjugated polymer–lipid hybrid NPs (**b**) unconjugated polymer–lipid hybrid NPs (Reused with permission from [49]. 2022 MDPI).

**Figure 2 polymers-14-04839-f002:**
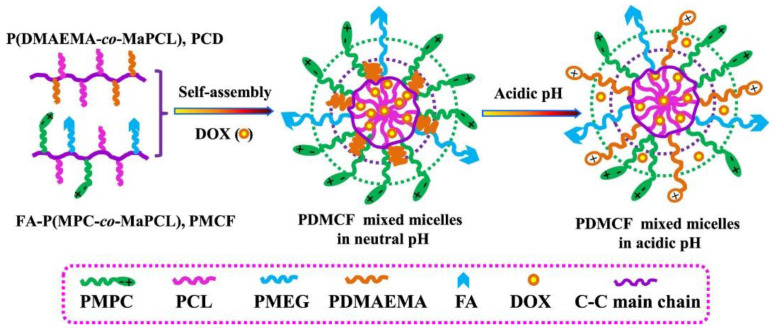
Schematic Illustration of the Formation of the Mixed Micelles and Their pH-Dependent DOX Release (Adapted and reused from [61], 2022, American Chemical Society).

**Figure 3 polymers-14-04839-f003:**
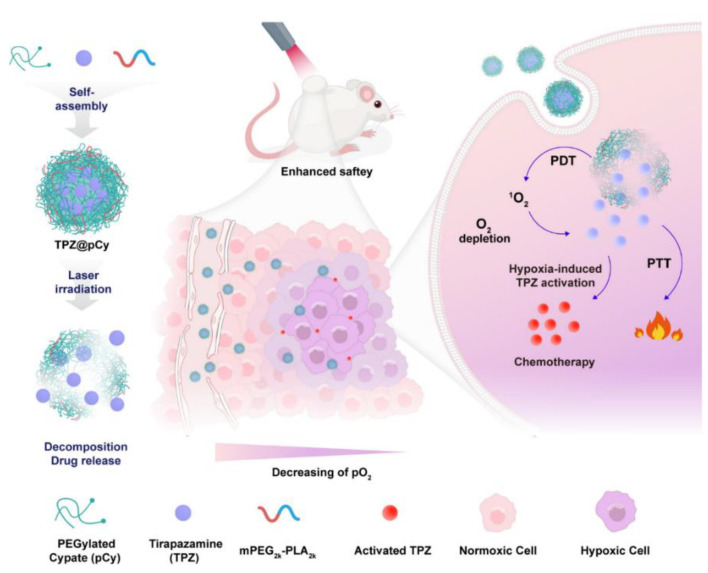
Schematic illustration of the fabrication process of TPZ@pCy and its mechanism of action for achieving controllable decomposition of nanocargo and delivery and activation of tirapazamine (Adapted and reused with permission from [65], MDPI 2022 under CCBY license).

**Figure 4 polymers-14-04839-f004:**
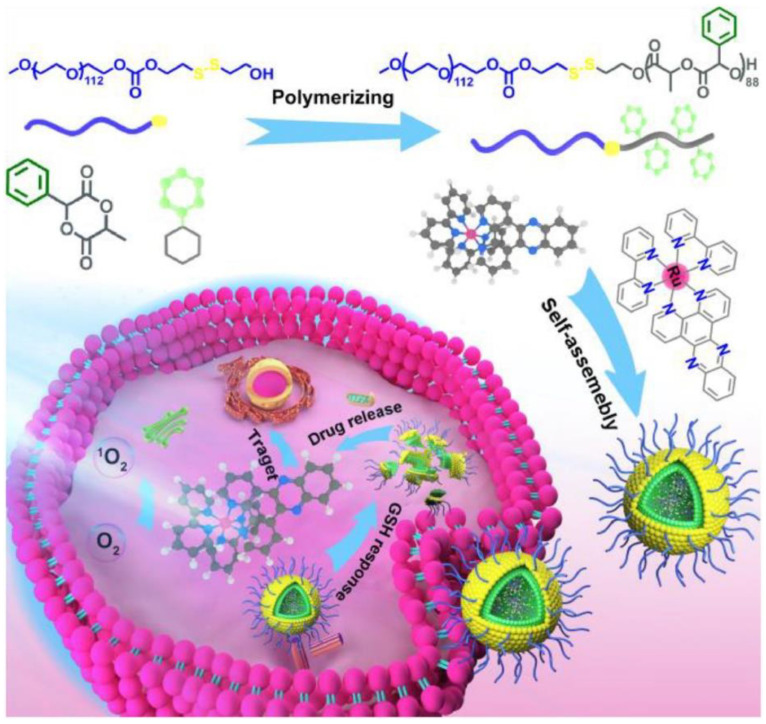
Schematic illustration of biodegradable Ru-containing polylactide MPEG-SS-PMLA@Ru micelles for enhanced Ru(II) polypyridyl complexes delivery and cancer phototherapy. (Adpated with copyright permission from [27]. 2022, American Chemical Society).

**Table 1 polymers-14-04839-t001:** Recent polymeric micelles prepared by self-assembly and their physicochemical characteristics and applications.

Polymer Backbone	Physicochemical Characteristics	Chemical Interactions	Applications	Ref.
Phenyl boronic acid conjugated mPEG-*b*-PCL	100 nm, spherical in shape	Hydrophobic, electrostatic and pendant interactions	micelles as DOX carriers for enhanced drug encapsulation and controlled drug release	[26]
Redox-responsive diblock copolymers (mPEG-SS-PMLA) of poly(ethylene glycol) and phenyl-functionalized poly(lactic acid) with disulfide bond as the linker are synthesized to prepare PLA-based micelles	131.0 nm, spherical particles	π-π interactions between Ru complex and polymer micelles	PDT against 4T1 tumor model by Ru-containing polylactide MPEG-SS-PMLA@Ru micelles	[27]
Imidazole-bearing block polymer	200 nm, spherical particles	Hydrogen bonding with DOX	DOX delivery to human ovarian adenocarcinoma	[28]
Hyaluronic acid (HA)-conjugated curcumin (Cur) and d-α-tocopherol acid polyethylene glycol succinate (TPGS)	66 nm, spherical particles	Possible hydrogen bonding, but there was no evidence justified	selective drug-carrying vehicles to deliver dasatinib (DAS) to HePG2 tumors	[29]
mPEG-PCL	~50 nm	Possible hydrogen bonding, but there was no evidence provided	metformin and Chrysin Delivery to Breast Cancer in Mice with improved efficacy	[30]
FA-PEG-PLGA	~80 nm	It could be hydrogen bonding interactions for the loading of two drugs	MCF-7/ADR xenograft tumor inhibition by the co-delivery of DOX and SIS3	[31]
PLA-PEG/SPIONS	-NA	Physical or hydrogen bonding could make such formulations	For delivery of galactose-targeted 19-O-triphenylmethylandrographolide (RSPP050) to HepG2 cells	[32]
PLA-PEG	NA	The process of drug encapsulation is mostly operated by the hydrophobic and hydrophilic interactions.	To improve the solubility of anticancer drug abemaciclib	[33]
Ru-containing polymer (poly(DCARu)) with two different therapeutics (the drug and the Ru complex) are rationally integrated and then conjugated to a MPEG-*b*-PMCC) containing hydrophilic poly(ethylene glycol) and cyano-functionalized polycarbonate	6–8 nm (TEM) and hydrodynamic size 22 nm	Metal-polymer complexation	Effective PDT against 4T1 tumor model	[34]
PEG-*b*-PAsp-BODIPY	110 nm, spherical shaped one	Covalent bonding	Effective PDT on C57BL/6J tumors in vivo	[35]
benzimidazole-terminated PHEMA-*g*-(PCL-BM) and β-CD-star-PMAA-b-PNIPAM	80 nm spherical particles	DOX was physically loaded during self-assembly	Dual responsive (pH and thermo) system for DOX release in to MCF-7 cells	[36]
PEG-PLA-IR-780	~118 nm	Physical encapsulation of IR-780	PDT on MCF-7 and MDA-MB-231 cell lines	[37]
PBA, F127 and ethanolamine were conjugated with poly(maleic anhydride)	~ 130 nm	Physical interactions	For improved targeted DOX delivery and was demonstrated on HepG2 tumor model	[38]

Foot notes: PDT, photo dynamic therapy; mPEG-*b*-PCL, Poly ethylene glycol-block-poly caprolactone; MCF-7/ADR, Adriamycin resistant MCF-7 breast cancer cells; mPEG-SS-PMLA, poly ethylene glycol-disulfide bonded-phenyl-functionalized poly(lactic acid); FA; folic acid; PEG, poly ethylene glycol; PLGA, poly(lactic-co-glycolic acid); BODIPY, Fluorinated Boron-Dipyrromethene dye; PBA- Phenyl boronic acid; PEG-*b*-PAsp, polyethylene glycol-b-poly(aspartic acid); PHEMA-*g*-(PCL-BM) and β-CD-star-PMAA-b-PNIPAM, poly(2-hydroxyethyl methacrylate)-graft-[polycaprolactone-benzimidazole: β-cyclodextrin-star-poly(methacrylic acid)-block-poly(N-isopropylacrylamide)].

**Table 2 polymers-14-04839-t002:** Major clinical trials reported with polymer micelles (all the information has been obtained from the national clinical trial website: https://clinicaltrials.gov/ct2/results?cond=&term=polymer+micelles&cntry=&state=&city=&dist=) Accessed on 8 November 2022.

No	Title	Status	Study Results	Conditions	Interventions	Locations	Clinical Trial No
1	A Study of Docetaxel Polymeric Micelles for Injection in Patients With Advanced Solid Tumors	Not yet recruiting	No Results Available	Advanced Solid Tumors	Drug: Docetaxel Polymeric Micelles for Injection	The First Affiliated Hospital of Bengbu Medical College, Bengbu, Anhui, China|The Forth Hospital of Hebei Medical University, Shijiazhuang, Hebei, China|Henan Cancer Hospital, Zhengzhou, Henan, China|The First Affiliated Hospital of Zhengzhou University, Zhengzhou, Henan, China|Hunan Cancer Hospital, Changsha, Hunan, China|Jiangxi Cancer Hospital, Nanchang, Jiangxi, China|Shandong Cancer Hospital, Jinan, Shandong, China|Shanghai East Hospital, Shanghai, Shanghai, China|Tianjin Medical University Cancer Institute&Hospital, Tianjin, Tianjin, China|Jinhua Municipal Hospital Medical Group, Jinhua, Zhejiang, China	NCT05254665
2	A Study to Evaluate ONM-100, an Intraoperative Fluorescence Imaging Agent for the Detection of Cancer	Completed	No Results Available	Breast Cancer|Head and Neck Squamous Cell Carcinoma|Colorectal Cancer|Prostate Cancer|Ovarian Cancer|Urothelial Carcinoma|Non-small Cell Lung Cancer	Drug: ONM-100	The University of Pennsylvania, Philadelphia, Pennsylvania, United States|The University of Texas Southwestern Medical Center, Dallas, Texas, United States|The University of Texas—M.D. Anderson Cancer Center, Houston, Texas, United States	NCT03735680
3	A Clinical Trial of Paclitaxel Loaded Polymeric Micelle in Patients With Taxane-Pretreated Recurrent Breast Cancer	Unknown status	No Results Available	Recurrent Breast Cancer	Drug: Paclitaxel loaded Polymeric micelle	Department of surgery, The Catholoic university of Korea, St. Mary’s hospital., Seoul, Korea, Republic of	NCT00912639
4	A Trial of Paclitaxel (GenexolÂ^®^) and Cisplatin Versus Paclitaxel Loaded Polymeric Micelle (Genexol-PMÂ^®^) and Cisplatin in Advanced Non Small Cell Lung Cancer	Completed	No Results Available	Non-Small Cell Lung Cancer	Drug: Paclitaxel (GenexolÂ^®^)|Drug: Paclitaxel loaded polymeric micelle (Genexol-PMÂ^®^)	Chungnam National University Hospital, Daejeon, Jung-gu, Korea, Republic of	NCT01023347
5	Paclitaxel-Loaded Polymeric Micelle and Carboplatin as First-Line Therapy in Treating Patients With Advanced Ovarian Cancer	Unknown status	No Results Available	Ovarian Cancer	Drug: carboplatin|Drug: paclitaxel-loaded polymeric micelle	Seoul National University Hospital, Seoul, Korea, Republic of|Yonsei Cancer Center at Yonsei University Medical Center, Seoul, Korea, Republic of|Samsung Medical Center, Seoul, Korea, Republic of|Asan Medical Center—University of Ulsan College of Medicine, Seoul, Korea, Republic of	NCT00886717
6	Docetaxel-polymeric Micelles(PM) and Oxaliplatin for Esophageal Carcinoma	Unknown status	No Results Available	Esophagus Squamous Cell Carcinoma (SCC)|Metastatic Cancer	Drug: Docetaxel-PM|Drug: Oxaliplatin	Dong-A University Hospital, Busan, Korea, Republic of	NCT03585673
7	Study of Genexol-PM in Patients With Advanced Urothelial Cancer Previously Treated With Gemcitabine and Platinum	Completed	No Results Available	Bladder Cancer|Ureter Cancer	Drug: Genexol PM	Asan Medical Center, Seoul, Korea, Republic of	NCT01426126
8	Clinical Investigation of the MiStent Drug Eluting Stent (DES) in Coronary Artery Disease	Completed	Has Results	Coronary Artery Disease	Device: MiStent DES|Device: Endeavor DES	Cardiovascular Center, Aalst, Belgium|Antwerp Hospital, ZNA Middelheim, Antwerp, Belgium|Brussels University Hospital, Brussels, Belgium|Ziekenhuis Oost-Limburg, Genk, Belgium|Virga Jesse Ziekenhuis, Hasselt, Belgium|KUL Cardiology Gasthuisberg, Leuven, Belgium|Jacques Cartier Hospital, Massy, France|Claude Galien Hospital, Quincy, France|Clinique Pasteur, Toulouse, France|OLV, Amsterdam, Netherlands|St. Antonius Ziekenhuis, Nieuwegein, Netherlands|TweeSteden Ziekenhuis, Tilburg, Netherlands|UMC Utrecht, Utrecht, Netherlands|Hospital Weezenlanden, Zwolle, Netherlands|Auckland City Hospital, Auckland, New Zealand|Mercy Angiography Unit, Auckland, New Zealand|Christchurch Hospital, Christchurch, New Zealand|Dunedin Hospital, Dunedin, New Zealand|Wellington Hospital, Wellington, New Zealand|Sahlgrenska University Hospital, Goteborg, Sweden|Orebro University Hospital, Orebro, Sweden|Royal Sussex Hosp, Brighton, United Kingdom|Papworth Hospital, Cambridge, United Kingdom|Guy’s & St. Thomas’, London, United Kingdom|Royal Brompton, London, United Kingdom|University Hospital South Manchester, Manchester, United Kingdom|Norfolk/Norwich UHosp, Norwich, United Kingdom|Southampton UHT, Southampton, United Kingdom	NCT01294748
9	First-In-Human Trial of the MiStent Drug-Eluting Stent (DES) in Coronary Artery Disease	Completed	Has Results	Coronary Artery Disease	Device: MiStent SES	St. Vincent’s Hospital Melbourne, Melbourne, Australia|Onze-Lieve-Vrouwziekenhuis Aalst (OLV Hospital), Aalst, Belgium|Ziekenhuis Oost-Limburg, Genk, Belgium|Auckland City Hospital, Auckland, New Zealand|Mercy Angiography Unit—Mercy Hospital, Aukland, New Zealand	NCT01247428
10	Study to Evaluate the Efficacy and Safety of Docetaxel Polymeric Micelle (PM) in Recurrent or Metastatic HNSCC	Unknown status	No Results Available	Head and Neck Squamous Cell Carcinoma	Drug: Docetaxel-PM	Samyang Biopharmaceuticals, Seoul, Korea, Republic of	NCT02639858
11	Paclitaxel in Treating Patients with Unresectable Locally Advanced or Metastatic Pancreatic Cancer	Completed	No Results Available	Pancreatic Cancer	Drug: paclitaxel-loaded polymeric micelle	Florida Cancer Specialists—Bonita Springs, Bonita Springs, Florida, United States|Midwest Cancer Research Group, Incorporated, Skokie, Illinois, United States|Louisiana Oncology Associates—Lafayette, Lafayette, Louisiana, United States|St. Vincent’s Comprehensive Cancer Center—Manhattan, New York, New York, United States|Southwest Regional Cancer Center—Central, Austin, Texas, United States	NCT00111904
12	Trial of MiStent Compared to Xience in Japan	Unknown status	No Results Available	Coronary (Artery); Disease	Device: MiStent (MT005) Coronary Artery Stent|Device: Xience Coronary Artery Stent	Iwaki Municipal Iwaki Kyoritsu Hospital, Iwaki-shi, Fukushima, Japan|Kansai Rosai Hospital, Amagasaki-shi, Hyogo, Japan|Tenyokai Central Hospital, Kagoshima-shi, Kagoshima, Japan|Kanto Rosai Hospital, Kawasaki-shi, Kanagawa, Japan|Sinkoga Hospital, Kurume-shi, Kurume-shi, Fukuoka, Japan|Omihachiman Community Medical Center, Omihachiman-shi, Shiga, Japan|Toho Univ.Ohashi Medical Center, Meguro-ku, Tokyo, Japan|Cardiovascular Institute Hospital, Minato-ku, Tokyo, Japan|Saiseikai Yokohama Tobu Hospital, Kanagawa, Yokohama, Japan|Shonan Kamakura General Hospital, Tokyo, Japan	NCT02972671
13	A Phase II Study of Weekly Genexol-PM in Patients with Hepatocelluar Carcinoma After Failure of Sorafenib	Terminated	No Results Available	Carcinoma, Hepatocellular	Drug: Genexol-PM	Gachon University Gil Medical Center, Incheon, Korea, Republic of|Samsung Medical Center, Seoul, Korea, Republic of	NCT03008512
14	A Phase II Trial of Genexol-PM and Gemcitabine in Patients with Advanced Non-small-cell Lung Cancer	Completed	No Results Available	Non-small Cell Lung Cancer	Drug: Genexol-PM/Gemcitabine	Gachon University Gil Medical Center, Incheon, Korea, Republic of	NCT01770795
15	Study Comparing the MiStent SES Versus the XIENCE EES Stent	Active, not recruiting	No Results Available	Coronary Stenosis	Device: MiStent|Device: XIENCE EES	Research Center Corbeil, Corbeil, France|Research Center Nimes, Nimes, France|Research Center Poitiers, Poitiers, France|Research Center Jena, Jena, Germany|Research Center Leipzig, Leipzig, Germany|Research Center Munster, Munster, Germany|Research Center Ulm, Ulm, Germany|Research Center Wiesbaden, Wiesbaden, Germany|Research Center Amersfoort, Amersfoort, Netherlands|Research Center Amsterdam, Amsterdam, Netherlands|Tergooi, Blaricum, Netherlands|Research Center Emmen, Emmen, Netherlands|Research Center Leeuwarden, Leeuwarden, Netherlands|Research Center Nijmegen, Nijmegen, Netherlands|Research Center Venlo, Venlo, Netherlands|Research Center Belchatow, Belchatow, Poland|Research Center Bielsko-Biala, Bielsko-Biala, Poland|Research center Chrzanow, Chrzanow, Poland|Research Center Tychy, Tychy, Poland|Research Center Zgierz, Zgierz, Poland	NCT02385279
16	Study of NC-6004 in Combination With 5-FU and Cetuximab in Patients with Head and Neck Cancer	Terminated	No Results Available	Head and Neck Neoplasms	Drug: NC-6004|Drug: Cetuximab|Drug: 5-FU	National Taiwan University Hospital, Taipei, Taiwan|Taipei Veterans General Hospital, Taipei, Taiwan|Chang Gung Memorial Hospital, Linkou Branch, Taoyuan, Taiwan	NCT02817113
17	Clinical Trial on the Efficacy and Safety of Sirolimus-Eluting Stent (MiStentÂ^®^ System)	Unknown status	No Results Available	Coronary Heart Disease	Device: MiStent|Device: TIVOLI	The Third Xiangya Hospital of Central South University, Changsha, Hunan, China|The First Affiliated Hospital of Baotou University, Baotou, Inner Mongolia, China|Inner Mongolia People’S Hospital, Hohhot, Inner Mongolia, China|The First Affiliated Hospital of Inner Mongolia Medical University, Hohhot, Inner Mongolia, China|The Second Affiliated Hospital of Nanchang University, Nanchang, Jiangxi, China|The First Affiliated Hospital of Xi’An Jiaotong University, Xi’an, Shaanxi, China|Sir Run Run Shaw Hospital School of Medicine, Zhejiang University, Hangzhou, Zhejiang, China|The General Hospital of Shenyang Military Region, Area Of Shenyang, China|Fu Wai Hospital, National Center for Cardiovascular Disease, Beijing, China|The First Hospital of Jilin University, Changchun, China|The Second Xiangya Hospital of Central South University, Changsha, China|The First Hospital of Lanzhou University, Lanzhou, China|Shanghai Ninth People’s Hospital, Shanghai, China|West China Hospital, Sichuan University, Sichuan, China|The Second Hospital of Shanxi Medical University, Taiyuan, China|TEDA International Cardiovascular Hospital, Tianjin, China	NCT02448524
18	Study to Evaluate the Efficacy and Safety of Genexol-PM Once a Week for Gynecologic Cancer	Unknown status	No Results Available	Gynecologic Cancer	Drug: Genexol-PM	Samyang Biopharmaceuticals, Seoul, Korea, Republic of	NCT02739529

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
