# Peer review of "Multifunctional Polymeric Micelles for Cancer Therapy"

_polymers, 2022, doi:10.3390/polym14224839_

Round 1
Reviewer 1 Report
This review highlights the progress in the development of multifunctional polymeric micelles in the field of cancer therapy, tumor imaging and theragnosis. The manuscript is comprehensive and clearly organized. I recommend this manuscript to be published after minor revisions.
1. Some minor mistakes, such as, there are two “and” in line 27. The literature format is not uniform, including punctuation and capitalization, such as 54,75,77
2. The first occurrence of PDT should be explained in Table 1 and the full name of PTT should be given.
3. In Table 1, all the preparation method of polymeric micelles is “self assembly”. It is better to classify polymer micelles according to the composition or preparation method, such as hydrophilic/hydrophobic, hydrogen bonding, electrostatic bonding or polyion complex micelle ......
4. Part 5, the title photo therapy should refer specifically to photothermal therapy and should be supplemented by more literature on stimulus response and photothermal therapy.
Author Response
the revised versions were marked in red color

Reviewer 2 Report
The manuscript provides an overview of the very current literature on the study of micelles in cancer therapy. I'd like to highlight, that the vast majority of references are articles from 2022. The review is very well structured and easy to read. Since the consideration of targeted and stimulus-responsive micelles is presented by the authors in separate paragraphs (4 and 5), I would recommend supplementing these paragraphs with summary tables (as authors have done in paragraph 2). This will greatly help the perception of these paragraphs.
Author Response
attached is the response to the comments
